# Dental Disease in Rabbits (*Oryctolagus cuniculus*) and Its Risk Factors—A Private Practice Study in the Metropolitan Region of Chile

**DOI:** 10.3390/ani13040676

**Published:** 2023-02-15

**Authors:** Tamara Palma-Medel, Daniela Marcone, Raúl Alegría-Morán

**Affiliations:** 1Departamento de Ciencias Clínicas, Facultad de Ciencias Veterinarias y Pecuarias, Universidad de Chile, Santa Rosa 11735, La Pintana, Santiago 8820808, Chile; 2Departamento de Medicina Preventiva Animal, Facultad de Ciencias Veterinarias y Pecuarias, Universidad de Chile, Santa Rosa 11735, La Pintana, Santiago 8820808, Chile; 3Escuela de Medicina Veterinaria, Facultad de Recursos Naturales y Medicina Veterinaria, Universidad Santo Tomás, Ejercito Libertador 146, Santiago 8370003, Chile

**Keywords:** rabbits, dental disease, ADD, malocclusion, radiography, risk factors

## Abstract

**Simple Summary:**

There has been an increase in domestic rabbit ownership in the recent years. The type of dentition of this exotic (non-traditional) animal pet, predisposes it to the development of Acquired Dental Disease (ADD), a disease that affects tooth quality, generates malocclusion, among other dental and non-dental consequences. The aim of this study was to assess the factors that modify the risk for ADD in domestic pet rabbits. For this, 1420 owned rabbits’ clinical records from a private practice from 2018 to 2021 were used, recording several variables, including the clinical signs at diagnosis. ADD was detected in 25.4% of the individuals, mostly on their cheek teeth. In addition, age and sex (male) were found to be significant risk factors for ADD. In contrast, a free lifestyle and consuming hay in the diet were protective factors. Understanding the factors that modify the risk of ADD increases the chances of prevention and helps to improve the education of owners and/or guardians of domestic rabbits.

**Abstract:**

Rabbits (*Oryctolagus cuniculus*) have elodont dentition, a characteristic that predisposes them to the development of Acquired Dental Disease (ADD), which is a multifactorial disease. The aim of this study was to assess the risk factors for ADD in domestic pet rabbits. To do this, a retrospective analysis of 1420 rabbits treated at a private practice during 2018–2021 was performed. For this, several variables were retrieved from clinical records, in addition to signology at the time of diagnosis. ADD was found on 25.4% of rabbits, mostly on their cheek teeth. In addition, age (OR = 1.029; 95% CI = 1.023–1.035; *p* < 0.001) and sex (male) (OR = 1.591; 95% CI = 1.226–2.064; *p* < 0.001) were found to be significant risk factors for ADD. In contrast, a free lifestyle (OR = 0.565; 95% CI = 0.362–0.882; *p* = 0.012) and consuming hay in the diet (OR = 0.323; 95% CI = 0.220–0.473; *p* < 0.001) were protective factors. In conclusion, ADD has a high prevalence and is usually underdiagnosed, highlighting the need for an exhaustive evaluation of patients during the clinical examination. This study improves our knowledge of ADD and its prevention.

## 1. Introduction

In recent years, there has been an increase in the ownership of exotic (non-traditional) animals as pets. Rabbits (*Oryctolagus cuniculus*) are some of the most popular domestic pets in European countries, being the third most frequent in the United Kingdom [1,2,3]; no similar information is available for Chile or other countries of South America, highlighting the importance of knowing the pathologies that may be associated with them. Likewise, exotic animal medicine has been updated with new diagnostic and therapeutic procedures in these species.

Rabbits belong to the order Lagomorpha, which are characterized by having four upper incisors, with a second pair located behind the primary incisors, called peg teeth. In addition, being heterodonts, they also have molars and premolars, which make up the functional unit in chewing, called molariforms or cheek teeth. That is why they present a dental formula of 28 teeth: 2 (I2/1, C0/0, PM3/2, M3/3) [4].

The oral cavity has the characteristic of presenting anisognathia, with the mandible being narrower than the maxilla, with a partial occlusion of 10% between the buccal edge of the mandibular teeth and the palatal edge of the maxillary teeth [5]. Its dental structure is composed of a clinical crown that protrudes from the dental alveolus, a reserve crown located subgingivally and intraosseously, and germinative tissue, without the presence of dental roots [4,6]. An important characteristic in rabbits is that they are elodont, meaning their teeth grow throughout their lives, maintaining a balance between growth and dental wear [2,5,7]. Among the factors that influence the speed of dental eruption is the axial load, since the growth rate doubles temporarily when there is no opposing force [5]. Another factor is the internal abrasive nature of the diet, where high-fiber foods produce better dental wear compared to mixed diets of muesli, pellets, or grains [8].

Dental disorders are one of the most common causes of veterinary consultation for these pets, followed by gastrointestinal, dermatological, ocular, and respiratory pathologies [9]. Most dental abnormalities in domestic rabbits are due to acquired dental disease (ADD), a progressive syndrome characterized by deteriorating tooth quality, acquired malocclusion, and elongation of reserve crowns.

The changes produced in the ADD, such as the overgrowth of the reserve crowns, generate secondary signs such as hyporexia and consequent weight loss in rabbits [10]. In addition, the nasolacrimal duct can become occluded, causing epiphora and dacryocystitis [4,10].

The growth of the clinical crowns of the cheek teeth generates tooth spurs that result in oral ulcers and stomatitis. In terminal stages, abscesses can occur due to penetration of the reserve crowns into the periosteum; in addition, the bones become osteopenic, susceptible to fractures [10,11].

The initial diagnosis is made based on the history and signs. To examine the oral cavity, a pediatric otoscope should be used in conjunction with a mouth gag to assess the length, position, structure, occlusion, and angulation of the teeth; this is best achieved under sedation [4,12]. For the definitive diagnosis, it is necessary to carry out complementary tests such as radiographic studies of the complete skull, which include projections: latero-lateral (LL), dorso-ventral (DV), right lateral oblique at 10–20° (OLD), left lateral oblique at 10–20° (OLI), ventro-dorsal (VD), rostro-caudal (RC), and intraoral (IO). However, the strict LL and DV views are the essential ones for diagnosis [13], since they allow lines to be drawn between specific anatomical points, allowing different degrees of dental disease to be diagnosed [14]. On the other hand, it is described in the literature that performing computed tomography (CT) allows for a more accurate diagnosis, generating 2D and 3D views for full skull reconstruction [4,15,16].

There are many studies on different etiologies in relation to the development of this syndrome; however, there is still no clarity on the exact cause. Among the most accepted theories is genetic predisposition with certain breeds, inadequate wear due to the type of diet, traumas, reproductive status, age, and metabolic bone disease [10].

This study aims to identify the prevalence of dental disease in rabbits of Chile, being the first study of this kind in the country, and also to identify the factors that are involved in the development of acquired dental disease in rabbits at a national level, along with identifying those most influential in its diagnosis.

## 2. Materials and Methods

### 2.1. Clinical Records

The data used in the study were obtained through a retrospective review of clinical records of rabbits that were assisted to a private veterinary clinic located in the commune of Ñuñoa in the Metropolitan Region, during the period between January 2018 and July 2021.

The total number of clinical records of patients admitted to the clinic during the study period was reviewed. Subsequently, those records of rabbits that were diagnosed with ADD were considered.

### 2.2. Inclusion Criteria

In the study, all rabbit patients admitted to the clinic during the mentioned study dates were included.

In addition, they were classified as patients with ADD when they were diagnosed with at least one of the following diagnostic methods:Oral cavity examination: patients who presented alterations in incisors and/or cheek teeth.Radiographic study: patients who underwent at least strict LL and DV views, presenting alteration of incisors and/or cheek teeth (clinical and reserve crowns).Computed tomography (CT): patients who underwent a cranial CT, presenting alteration of incisors and/or cheek teeth.

### 2.3. Exclusion Criteria

Those patients who, although part of the Lagomorpha order, did not belong to the study species (e.g., *Lepus europaeus*) were excluded; additionally, those that did not have a complete record in their clinical records, preventing the analysis of risk factors (such as oral cavity examination, age, weight, etc.), were also excluded.

### 2.4. Data Collection

The following data extracted from the clinical record of all the rabbits cared for during the established study period were recorded: record number, sex (male/female), age (measured in months), age was later categorized in newborn (up to 3 months), juveniles (from 3 to 12 months), adults (from 12 months to 7 years), and seniors (more than 7 years) considering prior information [17], weight (grams), body condition (BCS) (scale 1/5) [18], breed (dwarf/medium/large) based on previous data [17,19,20] (Appendix A), reproductive status (neutered/intact), type of housing (caged/free range pets/mixed), housing regimen (indoor/outdoor/mixed), consumption of hay (yes/no), consumption of vegetables (yes/no), consumption of fruit (yes/no), consumption of pellets (yes/no), diagnostic imaging (includes radiography and/or CT; yes /no), and clinical signs.

### 2.5. Data Analysis

The information obtained from the clinical records was entered into a database in an Excel^®^ spreadsheet (Version 16.68). The records were used to obtain the absolute frequency of patients with ADD based on the total population of rabbits admitted to the clinic during the study periods, also calculating the relative frequency with the value obtained.

In the characterization of ADD-positive patients, absolute and relative frequencies of each variable to be analyzed were used for the qualitative variables. For the quantitative variables, measures of central tendency (mean and median) and dispersion (standard deviation) were obtained, linked to the information extracted from the clinical records. The signs were grouped according to the affected systems (dermatological, gastrointestinal, ocular, neurological, and systemic).

Also, for this study, a multivariable logistic regression was performed to establish the predictive relationship between the categorical dependent variable (presence/absence of ADD) and the independent variables analyzed which were extracted from the clinical records. This analytical model was used as a result of the fact that the response variable (presence of dental disease) has dichotomous characteristics, that is, Y can have only two values, representing the absence (0) or the presence (1) of each of the agents studied (Y = 0 or Y = 1) [21,22,23,24]. All the individuals registered in the clinic were entered into the model, working with a control group (absence of ADD) and a study group (presence of ADD) to contrast the factors present in both, and to evaluate, through regression, which factors were expressed more in those individuals with dental disease, defining the latter as factors that modify the risk.

Initially, a univariable analysis was performed with the factors extracted from the medical records, where a liberal *p* value criterion (*p* < 0.15) was considered to select those variables that entered the multivariate model. Spearman’s correlation tests, Chi-square, and Fisher’s exact test were performed to evaluate collinearity and/or association between the variables that exceeded the cut-off [24]. A stepwise backward elimination approach was used for the construction of the final model, using the log Likelihood Ratio Test (LRT) for the comparison between models, selecting those models that present lower LRT values, removing those non-significant variables (*p* > 0.05) [22]. The goodness of fit of the model was evaluated using the Holmer–Lemeshow Test [23]; McFadden’s pseudo-R^2^ value was calculated [25].

All analyses were performed using the statistical program R [26] and R Studio [27] in the latest versions available (versions 4.2.2 and 2022.12.0+353 respectively), allowing the identification and quantification of those variables that are factors which modify the risk of presenting with dental disease in the analyzed group.

## 3. Results

### 3.1. Frequency of Rabbits Diagnosed with ADD

Between 2018 and 2021, a total of 1420 rabbits were registered, of which 361 were diagnosed with ADD, obtaining a relative frequency of 0.254; so, 25.4% of the population presented this disease at some point in their life.

### 3.2. Characterization of Patients with ADD

Of the 361 rabbits diagnosed with ADD, 75 (21%) rabbits presented a radiographic study, and the rest were clinically diagnosed.

A mean age of 31 months was obtained, in which 30% were between neonates and juveniles, 64% were adults, and 6% corresponded to the elderly group. Regarding weight, a mean of 2147 g was obtained; further details of the descriptive statistic can be observed in Table 1. In addition, most of them were individuals of medium race (68%), with a BCS 3/5. The disease was observed with a higher percentage in the cheek teeth (55%), compared to the incisors (26%) and all the teeth (19%). Additionally, it was observed in a higher proportion of males compared to females in the present study (62% vs. 38%, respectively). Regarding reproductive status, 18% of the rabbits were sterilized and 82% were intact rabbits at the time of diagnosis. Regarding their lifestyle, it is reported that 10% lived in a cage, 54% lived as free-range pets, and 36% were kept in a mixed system. In addition, it was recorded that the majority lived in an indoor system (74%) compared to an outdoor system (11%) and the mixed system (15%). Finally, in the diet records, 80% consumed hay, 96% pellets, 78% vegetables, and 71% fruit, being non-exclusive among them (Appendix A).

The signs of the patients who presented ADD were recorded (Appendix A), and were then grouped according to the affected system. Forty-two percent (42%) of the rabbits did not present signs associated with the disease, 15% presented ocular signs (O), 11% dermatological signs (D), 8% gastrointestinal signs (G), and <1% presented only neurological signs (N). In addition, 20% presented mixed signs (8% G + O; 4% G + D; 1% G + N; 5% O + D; 1% O + N; and 1% D + N), and 4 % recorded systemic signs (O + D + G + N).

### 3.3. ADD Risk Factors

Prior to the final model, collinearity tests were performed, where age turned out to have a high connection with weight. In addition, interactions with biological and epidemiological coherence between the variables to be evaluated are included, without having statistical significance (*p* > 0.05), and were left out of the final analysis.

The significant variables in the model that explain the phenomenon of presenting with dental disease in rabbits from the Metropolitan Region were age, which acts as a factor that increases the risk of presenting ADD (odds ratio (OR) = 1.029; CI-95%: 1.023–1.035; *p* < 0.001), and sex, where being male increases the risk compared to females (OR = 1.591; 95% CI: 1.226–2.064; *p* < 0.001). On the contrary, two variables were significant, behaving as protection factors, meaning they reduce the risk of presenting with ADD. These were being free range pets (not caged) (OR = 0.565; CI-95%: 0.362–0.882; *p* = 0.01) and consuming hay as part of their diet (OR = 0.323; 95% CI: 0.224–0.473; *p* < 0.001) (Table 2). The model shows a good adjustment at the Hosmer–Lemeshow test (*p* = 0.136), and a McFadden’s pseudo-R^2^ value of 0.166.

## 4. Discussion

Dental pathologies are of great importance in elodont species, affecting other systems and the life expectancy of patients. This study presented a prevalence of ADD of 25.4% (n = 361), which correlates with previous studies carried out in the United Kingdom and Finland where they obtained a prevalence that varies between 30% [28,29] and 40%. In Mexico, a 42.8% prevalence was reported for ADD [30] and this was close to 30% in the case of ADD reports in Perú [31], being the only reports for Central and South America to-date.

A higher prevalence of the disease was obtained in the location of cheek teeth, which is associated with their chewing mechanism, since this is associated with grinding cycles that can be more easily altered in dental wear if rabbits do not consume an adequate diet. ADD follows a pattern in the cheek teeth during the intermediate stage, where spurs begin to form on the second, third, or fourth lower cheek teeth according to Harcourt-Brown and Chitty [10]. Similarly, in a study by Artiles, et al. [32], 100 cases of rabbits with dental disease were analyzed by CT, in which greater alterations were observed in the cheek teeth (curvatures, dental elongation, etc.) compared to the incisors.

When evaluating the signology of the disease, a higher percentage was asymptomatic at the time of diagnosis. This could be explained by rabbits being prey animals, meaning they hide signs of illness, which is why pathologies often develop long before they present to the veterinary clinic. In this context, a health survey of 102 domestic rabbits in the UK by Mullan and Main [28] revealed that 30 rabbits had dental disease and only 6 of their owners were aware of the problem; that is, owners often are unaware of their rabbits’ dental problems.

In addition, the ocular, dermatological, and gastrointestinal systems were the most affected systems secondary to ADD in the present study, which may be associated with overgrowth of initial reserve crowns during the disease, the impediment of rabbits to groom themselves, and the difficulty of feeding due to dental alteration or pain, respectively. Similarly, in the study by Artiles, et al. [32], the most common clinical signs at the time of evaluation included epiphora, abscesses, anorexia or hyporexia, and runny nose. Likewise, in the study by Mäkitaipale, et al. [1], ocular discharge was strongly associated with ADD diagnosed during physical examination.

In this report, most of the rabbits corresponded to the adult age group (64%), with age being a risk factor for ADD. This can be explained as time must elapse during which different factors (diet, lifestyle, environment, etc.) associated with ADD influence it. Therefore, at older ages, this longer exposure time results in a deterioration in dental quality. Similarly, when evaluating the risk factors of ADD, age turned out to be the only significant variable in the study by Artiles, et al. [32], which was associated with the fact that this disease is progressive in time. These results are in contrast to those described by García and Maldonado [30], where in addition to adult rabbits, the juvenile age group was also part of the majority of rabbits diagnosed with ADD through radiographic findings; this could be associated, since at juvenile stages, there is a high demand for calcium for growth, leading to dental alterations such as enamel defects due to a lower concentration of this mineral [33].

Patients in this study had an average weight of 2147 g with a BCS 3/5 at the clinical examination. Similar results were observed previously with an average weight of 2380 g with a BCS between 2.5 and 3.5 (considering intermediate values), where a higher percentage of the total number of rabbits presented with ADD [28]. On the other hand, other studies report poor body condition and anorexia linked to the difficulty and pain when consuming food, associated with ADD [34,35]. This difference could be explained because most of the rabbits in this study were asymptomatic at the time of diagnosis, so they did not have difficulty in eating, and therefore had a normal body score.

Regarding breed, most of the rabbits in the study belonged to the medium breed; however, it did not turn out to be a significant factor for ADD, unlike other investigations where breed had an implication in the development of the disease. According to the literature, rabbits of dwarf and small breeds are more frequently affected by dental diseases than animals of larger breeds [28,36,37], given the brachycephalic trait, the inherited mandibular prognathism and maxillary brachygnathism in dwarf breeds leading to incisor malocclusion [29,37,38,39]. In the present study, the breeds were evaluated according to the dwarf, medium, and large categories, since in most of the rabbits the specific information was not available due to the tutor’s ignorance, and because the majority turned out to be mixed breeds. At the national level, rabbit farms are not certified or supervised, unlike other countries that have breeder associations, which recognize different breeds according to a standardized guide. Likewise, at a clinical level, it has been observed over the years that most of the specimens turn out to be mixed breeds, which may influence the difference in the prevalence of ADD in rabbit breeds compared to other countries.

When evaluating sex in the present study, the disease occurred mainly in males compared to females, which turned out to be significant as an ADD risk factor, being a controversial cause among the different studies. Several studies reported an increased risk of ADD in males when compared to female rabbits [34,40]. This may be associated with a hormonal factor, since estrogens in females raise calcium serum concentration, participating in its intestinal absorption and influencing the conversion to active vitamin D3 [41], therefore, helping the formation of dentin and enamel. It is also described in the literature that there is a greater susceptibility to the development of osteoporosis and dental alteration in castrated females (no estrogen) due to calcium deficiency [42,43]. It is postulated that the lack of this hormone in males could be associated with the development of dental disease; however, other authors suggest that sex does not seem to be influential in ADD [1,30,44]. In relation to this area, most of the analyzed patients were in an intact state, probably associated with the fact that most of them were rabbits attending their first clinical consultation. In addition, a sex ADD association could be linked to the development of overgrown molars in male compared to female rabbits, partially related to sexual dimorphism [34], also linked to the effect of androgens [45].

Regarding lifestyle, most of the analyzed rabbits were free ranging pets in an indoor system, which is associated with the ownership and care of this species, since they tend to be more delicate than traditional pets. Keeping rabbits as free range pets, either indoors or outdoors, was found to be significant as a protective factor for ADD, which may be due to the fact that keeping them as free ranging pets allows more options to gnaw, and decreases the chances of traumatic injuries associated with being caged, such as fractured incisors from being caught between the cage bars [11,16,42]. Cage chewing has been reported to occur as stereotypy in domestic and laboratory rabbits [28,46]. In addition, animals that graze freely choose different types of food that will allow for optimal tooth wear [47,48,49,50]. As a result, it is important to highlight that sun exposure, ultraviolet B (UVB) radiation, plays an important role in the synthesis and activation of vitamin D when animals do not acquire it through their diet, potentially leading to dental alteration in animals with an indoor lifestyle without sunlight or UVB exposure [51,52].

In this report, most of the rabbits that presented with the disease consumed hay, pellets, fruits, and vegetables in their diet. However, only hay was found to be significant for the disease, acting as a protective factor. In various studies, it has been observed that diet influences dental growth and wear, where consuming hay promotes adequate movement and wear during chewing, reducing dental alterations compared to those who do not consume it [18,44,53]. Similar results were obtained in the study by Meredith, et al. [2] when comparing the effect of different types of diets on the shape, growth, and dental wear, in which it was observed that rabbits without access to hay presented higher dental alterations. On the other hand, it has been suggested that pellet consumption is a risk factor for ADD because it produces changes in chewing patterns [8,28], while in the study by Müller, et al. [7], it was observed that the pellet with high fiber content does not increase the risk of dental diseases in the short term. This discrepancy can be explained by the quality of food consumed and the exposure time that leads to changes in tooth structure. In the present study, the proportions and quality of each component in the diet were not evaluated, which may explain the difference in the results according to the literature.

## 5. Conclusions

According to our study, ADD is highly prevalent in rabbits of the Metropolitan Region, Chile, and due to the progress and mechanism of the disease, it also affects different systems (ocular, dermatological, and gastrointestinal) which is why it is not always detected by owners, making it difficult to diagnose. This study therefore shows that the disease is often underdiagnosed, since most rabbits arrive at the clinic with advanced stages of the disease, because the first ADD changes are not evident. Likewise, this study demonstrates the relevance of carrying out an exhaustive analysis, comprehensively evaluating the patient in conjunction with carrying out complementary tests that allow reaching an early diagnosis.

Multiple causes have been investigated for the presentation of ADD. In this study, age and being male turned out to be risk factors for dental disease, while maintaining a free-range type of housing and consuming hay within the diet decreased the disease presentation. In other studies, breed and pellet consumption also influenced the development of ADD, so it is considered important to evaluate these data in new research at the national level, through a better collection of information on each variable, evaluating the proportion, quality, and intensity of dietary components to establish better prevention measures.

Finally, and as a conclusion, this work is the first of its kind in Chile and lays the foundations for the study of dental disease in rabbits in the country, knowing its development, its risk factors, and diagnosis. However, being a retrospective analysis, it has the limitations of veracity and lack of information, which is why it presents necessary challenges for future studies. These should focus on the follow-up of patients from the beginning to analyze, in more detail, the factors that influence disease presentation.

## Figures and Tables

**Table 1 animals-13-00676-t001:** Quantitative summary measures in patients diagnosed with ADD in a private veterinary clinic between the years 2018 and 2021.

Measures	Age (Months)	Weight (g)
n ^1^	361	361
Mean	30.720	2147.044
S.D ^2^	27.107	751.495
Var ^3^	732.728	563,179.920
VC ^4^	88.237	35.001
Min ^5^	1	130
Max ^6^	156	4490
Median	23	2160
Asymmetry	1.379	0.047
Kurtosis	2.128	0.034

n ^1^ = Sample Size, S.D ^2^ = Standard Deviation, Var ^3^ = Variance, VC ^4^ = Variation Coefficient, Min ^5^ = Minimum value, Max ^6^ = Maximum value.

**Table 2 animals-13-00676-t002:** Results of the statistical model of ADD risk factors in rabbits who attended a private veterinary clinic between the years 2018 and 2021.

Variable	Category	*p*-Value	OR	CI 95%	
				Lower	Upper
(Intercept)		0.048	0.583	0.342	0.994
Age		**<0.001**	1.029	1.023	1.035
Sex	F ^1^	reference			
	M ^2^	**<0.001**	1.591	1.226	2.064
Type of housing	C ^3^	reference			
	Mx ^4^	0.241	0.759	0.479	1.204
	FRP ^5^	**0.012**	0.565	0.362	0.882
Hay	0 ^6^	reference			
	1 ^7^	**<0.001**	0.323	0.220	0.473

F ^1^ = Female, M ^2^ = Male, C ^3^ = Cage, Mx ^4^ = Mixed, FRP ^5^ = Free range, 0 ^6^ = Absent, 1 ^7^ = Present. Results of *p*-value bold highlighted indicate statistically significant values (*p* < 0.05).

## Data Availability

The data presented in this study are available on request from the corresponding authors. The data are not publicly available because they are part of an ongoing project not yet published.

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
