# Peer review of "Dental Disease in Rabbits (Oryctolagus cuniculus) and Its Risk Factors—A Private Practice Study in the Metropolitan Region of Chile"

_animals, 2023, doi:10.3390/ani13040676_

Round 1

Reviewer 1 Report

Overall

* Impact of this article is weak. It is difficult to understand the point of argument.

* Too long sentence but the contents are poor.

* In particular, the Introduction and Discussion are too long and talkative. Authors have to state more clearly and directly.

* What are the new findings in this article?

* All tables are no good. Difficult to understand. Please revise.

* The results that ADD was found on 30% of rabbits are same as previous report. What is the originality of this article?

Details

L 278-283. Discussion between ADD and body condition is insufficient.

L 302-319. Discussion of gender is also difficult to understand.

L 255-263. I cannot understand this discussion. If authors considered the overgrowth is associated with ADD, authors should discuss with body condition score.

* In abstract (L 28-29.), authors stated that ADD is classified 5 degrees. But there is no description in Materials & Methods and Results. Why?

Author Response

- Reviewer Nº1:

We greatly appreciate your comments and suggestions, all of them were included in the new version of the manuscript.

Overall

Q1.- Impact of this article is weak. It is difficult to understand the point of argument.

R: We very much appreciate the comment. Rabbits (Oryctolagus cuniculus) are the third most frequent pet in Europe and America (Meredith et al., 2015; PDSA, 2022). In Chile there is no exact census, but an increase of the ownership of this pet can be seen due to the increase in rabbits visiting veterinary clinics daily. So, the information, both for clinicians and for owners/tutors, is scarce. Furthermore, in Chile there are no scientifically based studies that address this problem. Our aim was to generate information on the positivity rate, plus to risk factors determination in domestic rabbits from Chile. Several changes have been incorporated into the manuscript to increase the understanding of the point of argument.

  • Meredith, A.L.; Prebble, J.L.; Shaw, D.J. Impact of diet on incisor growth and attrition and the development of dental disease in pet rabbits. Journal of Small Animal Practice 2015, 56, 377-382, doi:https://doi.org/10.1111/jsap.12346.
  • Animal Wellbeing Report. 2022; https://www.pdsa.org.uk/what-we-do/pdsa-animal-wellbeing-report/past-reports.

Q2.- Too long sentence but the contents are poor.

R: We very much appreciate the comment. Several changes have been incorporated into the manuscript to reduce sentence length.

Q3.- In particular, the Introduction and Discussion are too long and talkative. Authors have to state more clearly and directly.

R: We very much appreciate the comment. Changes have been made on the Introduction and Discussion sections to shorten them and make them more concise and clearer. Changes can be observed at lines 44 – 52, 58 - 59, 66, 72, 77, 101 – 102, 103 in the Introduction, and 306 - 310, 335, 350, 361-363, 366, 396, 401 - 405, 420 - 425, 458, 459 and 462 in the Discussion.

Q4.- What are the new findings in this article?

R: We very much appreciate the comment. The information and results presented in this article are new to our country and many other South American countries, since it’s the first study of its kind. Changes have been made on Introduction and Conclusion to make this situation more explicit. Changes can be observed at lines 44 – 53 and 101 – 102 (Introduction), and 486 (Conclusion).

Introduction: (...) This study aims to identify the prevalence of dental disease in rabbits of Chile, being the first study of this kind in the country. And also, it seeks to identify the factors that are involved in the development of acquired dental disease in rabbits at a national level, along with those most influential in its diagnosis (...).

Conclusion: (...) Finally, and as a conclusion, this work is the first of its kind in Chile and lays the foundations for the study of dental disease in rabbits in the country, knowing its development, its risk factors and diagnosis (...).

Q5.- All tables are no good. Difficult to understand. Please revise.

R: We very much appreciate the comment. Tables have been revised and are in accordance with the format of the journal, considering that one of the authors already presents previous publications in this journal. Additional efforts have been made to make tables easier to understand. Table 1 was removed from the manuscript and incorporated to supplementary materials, tables 2 and 3 where also modified (lines 248 - 251, and 299 - 302).

Q6.- The results that ADD was found on 30% of rabbits are same as previous report. What is the originality of this article?

R: We very much appreciate the comment. Indeed, the prevalence of ADD found in this study (25.4%) is similar to other international reports, in Latin America a prevalence of 29.5% was determined in Peru (Peña, 2022) and 42.8% in Mexico (García and Maldonado, 2020). This research corresponds to the first of its kind in Chile, since there are no current records in the country on the level of ADD in populations of domestic rabbits kept as pets. On the other hand, the aim of this study was not only to estimate the frequency (prevalence/positivity rate) of ADD but also to identify factors that modify the risk of presentation. This will make possible to contribute to future development of evidence-based medicine protocols for approaching patients and to help clinicians of exotic species to educate tutors on how to reduce the risk of ADD presentation. Finally, similar studies have analyzed small populations of rabbits (around 170 individuals), while this study involves more than 1,400 patients and therefore provides a more robust estimate of ADD in domestic rabbits kept as pets.

  • García, M.C.; Maldonado, R.I. Prevalencia y hallazgos radiográficos en conejos, cuyos y chinchillas diagnosticados con maloclusión en el Hospital Veterinario de Especialidades en Fauna Silvestre y Etología Clínica de la UNAM, México. Revista de la Facultad de Medicina Veterinaria y de Zootecnia 2020, 67, 17-32, doi:https://doi.org/10.15446/rfmvz.v67n1.87676.
  • Peña, K. Factores que influyen a la maloclusión dental de los conejos (Oryctolagus cuniculus) en la Clínica Veterinaria Vale Vet en el Distrito de Surquillo – 2021. Universidad Nacional Hermilio Valdizán, Huánuco, Perú, 2022. https://hdl.handle.net/20.500.13080/7541

Details

Q7.- L 278-283. Discussion between ADD and body condition is insufficient.

R: We very much appreciate the comment. Authors agree on the comment and have incorporated further details. The manuscript has been changed to increase the depth of the discussion in this aspect (lines 361 - 363).

“ (…) Patients in this study had an average weight of 2,147g with a BCS 3/5 at the clinical examination. Similar results were observed previously with an average weight of 2,380 g with a BCS between 2.5 and 3.5 (considering intermediate values), where a higher percentage of the total number of rabbits presented ADD [27]. On the other hand, other studies report poor body condition and anorexia linked to the difficulty and pain when consuming food, associated to ADD [33,34]. This difference could be explained because most of the rabbits in this study were asymptomatic at the time of diagnosis, so they didn’t have difficulty in eating and therefore, had a normal body score. (…)”

Q8.- L 302-319. Discussion of gender is also difficult to understand.

R: We very much appreciate the comment. Authors agree on the comment and have incorporated further details. The writing of paragraph has been changed to make it easier to understand (lines 401 - 402, and 403 - 405).

“(...) This may be associated with a hormonal factor, since estrogens in females raise calcium serum concentration, participating in its intestinal absorption and influencing the conversion to active vitamin D3 [40]. Therefore, helping the formation of dentin and enamel. It is also described in the literature as a greater susceptibility to the development of osteoporosis and dental alteration in castrated females (no estrogen) due to calcium deficiency [41,42]. It is postulated that the lack of this hormone in males could be associated with the development of dental disease; however, other authors suggest that sex does not seem to be influential in ADD [1,29,43]. In relation to this area, most of the analyzed patients were in an intact state, probably associated with the fact that most of them were rabbits attending their first clinical consultation. (...)” 

Q9.- L 255-263. I cannot understand this discussion. If authors considered the overgrowth is associated with ADD, authors should discuss with body condition score.

R: We very much appreciate the comment. If the reviewer is referring to the paragraph starting with “(…) In addition, the ocular, dermatological and gastrointestinal systems were the most affected systems secondary to ADD in the present study, which may be associated with overgrowth of initial reserve crowns during the disease, (…)”. Authors emphasize that this topic is explained at lines 77 - 80 (Introduction) and was further explained at the changes incorporated at Q7 of this response letter (lines 361 - 363).

Q10.- In abstract (L 28-29.), authors stated that ADD is classified 5 degrees. But there is no description in Materials & Methods and Results. Why?

R: We very much appreciate the comment. Authors agree with the reviewer. The content related to the 5 degrees of ADD was removed from the manuscript, considering that to determine the degree of ADD, complementary radiographic examinations are required, which were not present in all the individuals that were part of the present study due to the recent incorporation of this complementary exams into the routine protocol for ADD patients in Chile.

Reviewer 2 Report

The Authors conducted a study to assess the risk factors for acquired dental disease in domestic pet rabbits (Oryctolagus cuniculus). A retrospective analysis of 1,420 rabbits was performed using clinical records from a private veterinary clinic. The results show that age and sex are considered significant risk factors for acquired dental disease while, in contrast, free lifestyle and the ingestion of hay in the diet are protective factors. This is an important contribution for veterinary clinics and the manuscript is suitable for this journal. Overall, the manuscript is noticeably clear, well-structured and well referenced.

The manuscript can be accepted for publication into Animals. However, there are few typos/ errors that need attention:

Minor comments:

*Lines 28 and 71. Change “5” to “five”. Numbers under 10 must be spelt out, except for: measurements with a unit (24 kg); age (8 years old), or lists with other numbers (13 dogs, 3 cats, 6 mouses). Please, check the text main text.

*Line 131. Change “age1” to “age”.

*Table 1. Abbreviations for the same term do not match: “Dn” for the term within table vs “D” for the footnotes of the table.

*Line 164. I assume that the p-value is 0.05 and not 0.15 as shown in the text.

*Line 178. Change “1420” to “1,420”.

*Line 179. The frequency is 0.254 and not 0.256 as indicated in the text. In this way, 25% of the population of domestic rabbits presented acquired dental diseases and not 30% as indicated in the text. Please, change the value “30%” to “25%” in lines 21 (simple summary), 31 (abstract), 179 and 233.

*Line 187. Change “2147” to “2,147”.

*Table 2. All abbreviations must be described, including, n, CV, Min and Max.

*Line 204. Do not start a sentence with a number. Change “42%” to “forty-two percent”

*Line 218. The variable “gender” is used as synonym of “sex”. Sex and gender are not synonyms. “Sex” is a biological term. It refers to the biological differences between males and females, such as the genitalia and genetic differences. “Gender” is more difficult to define, but it can refer (in humans not in animals) to the role of a male or female in society (gender role) or an individual’s concept of themselves (gender identity). Please, change these terms as they are not interchangeable.

*A personal tip for authors. In the conclusions section, the Authors give certain indications of variables to be considered for future research (for example: proportion, quality, and intensity of dietary components). Being a study based on domestic rabbits, their diet depends on what their owners provide them. In this context, in addition to food, it would also be interesting to analyze the drinking water of these animals. That is, if they drink bottled water or tap water. Artificial fluoridation of public drinking water has been the most efficient measure for the collective prophylaxis of dental caries in humans, so if the drinking water of these rabbits is tap water (provided by the owners), they could be partially “protected” against dental diseases such as caries. Although the opposite case could also occur, and excessive fluoride consumption could cause dental fluorosis.

Author Response

- Reviewer Nº2:

We greatly appreciate your comments and suggestions, all of them were included in the new version of the manuscript.

The Authors conducted a study to assess the risk factors for acquired dental disease in domestic pet rabbits (Oryctolagus cuniculus). A retrospective analysis of 1,420 rabbits was performed using clinical records from a private veterinary clinic. The results show that age and sex are considered significant risk factors for acquired dental disease while, in contrast, free lifestyle and the ingestion of hay in the diet are protective factors. This is an important contribution for veterinary clinics and the manuscript is suitable for this journal. Overall, the manuscript is noticeably clear, well-structured and well referenced.

The manuscript can be accepted for publication into Animals. However, there are few typos/ errors that need attention:

Minor comments:

Q1.- Lines 28 and 71. Change “5” to “five”. Numbers under 10 must be spelt out, except for: measurements with a unit (24 kg); age (8 years old), or lists with other numbers (13 dogs, 3 cats, 6 mouses). Please, check the text main text.

R: We very much appreciate the suggestion. The topic of the different degrees of ADD was removed from the manuscript, due to low sample size with the complementary radiographic examinations required, which were not present in all the individuals that were part of the present study due to the recent incorporation of this complementary exams into the routine protocol for ADD diagnostic in patients in Chile.

Q2.- Line 131. Change “age1” to “age”.

R: We very much appreciate the comment. Change was incorporated at line 154

Q3.- Table 1. Abbreviations for the same term do not match: “Dn” for the term within table vs “D” for the footnotes of the table.

R: We very much appreciate the comment. Change was incorporated at Supplementary material 1.

Q4.- Line 164. I assume that the p-value is 0.05 and not 0.15 as shown in the text.

R: We very much appreciate the comment. The text is correct, ate the univariable analysis a liberal criteria was used,this criteria indicates that any variable that presents a p < 0.15 at the univariable step, can be selected for the multivariable analysis (Dohoo et al., 2012).

  • Dohoo, I.; Martin, W.; Stryhn, H. Methods in Epidemiologic Research, First ed.; 1st ed. VER Inc. Charlottetown, PEI, Canada.: 2012; pp. 890.

Q5.- Line 178. Change “1420” to “1,420”.

R: We very much appreciate the comment. Change was incorporated at line 220

Q6.- Line 179. The frequency is 0.254 and not 0.256 as indicated in the text. In this way, 25% of the population of domestic rabbits presented acquired dental diseases and not 30% as indicated in the text. Please, change the value “30%” to “25%” in lines 21 (simple summary), 31 (abstract), 179 and 233.

R: We very much appreciate the comment. Changes were incorporated at lines 22, 32, 221, and 306.

Q7.- Line 187. Change “2147” to “2,147”.

R: We very much appreciate the comment. Change was incorporated at lines 248 - 251.

Q8.- Table 2. All abbreviations must be described, including, n, CV, Min and Max.

R: We very much appreciate the comment. Change was incorporated at lines 250 - 251.

Q9.- Line 204. Do not start a sentence with a number. Change “42%” to “forty-two percent”

R: We very much appreciate the comment. Changes were incorporated at line 254.

Q10.- Line 218. The variable “gender” is used as synonym of “sex”. Sex and gender are not synonyms. “Sex” is a biological term. It refers to the biological differences between males and females, such as the genitalia and genetic differences. “Gender” is more difficult to define, but it can refer (in humans not in animals) to the role of a male or female in society (gender role) or an individual’s concept of themselves (gender identity). Please, change these terms as they are not interchangeable.

R: We very much appreciate the comment. Authors agree with the suggestion. Change was incorporated at line 269.

Q11.- A personal tip for authors. In the conclusions section, the Authors give certain indications of variables to be considered for future research (for example: proportion, quality, and intensity of dietary components). Being a study based on domestic rabbits, their diet depends on what their owners provide them. In this context, in addition to food, it would also be interesting to analyze the drinking water of these animals. That is, if they drink bottled water or tap water. Artificial fluoridation of public drinking water has been the most efficient measure for the collective prophylaxis of dental caries in humans, so if the drinking water of these rabbits is tap water (provided by the owners), they could be partially “protected” against dental diseases such as caries. Although the opposite case could also occur, and excessive fluoride consumption could cause dental fluorosis.

R: The authors greatly appreciate this comment. Not only fluoridation is relevant, but chlorination will also be different depending on the water suppliers, and water-hardness will be different. At least in Chile, most rabbit tutors would use bottled water, it would be interesting to evaluate this situation in future research.

Reviewer 3 Report

Dear authors,

The study, as mentioned in the text, is related to a certain region of Chile, which in my opinion should be referred to in the title and more explicitly in the text, given that the dissemination of the article after publication will be worldwide and not national.

Title: Dental disease in rabbits (Oryctolagus cuniculus)  - a private practice study in the Metropolitan Region of Chile.

Lines 50 to 51 - In addition, being heterodontos, they also have molars and premolars, which make up the functional unit in chewing, called molariforms  or  cheek teeth

Line 128 - 2.4 Data Collection

Line 132 -  age1 (measured

Do you have the bibliographic reference of the body condition scale?

statistical p in italics

Line 185 - A mean age of 31 months was obtained, in which 30% were between neonates and juveniles, 64% were adults, and 6% corresponded to the elderly group.

Authors must previously, in the material and methods section, refer to the considered age groups.

Line 233 and 301 - prevalence and incidence are not synonymous. In this study, prevalence was studied, not incidence.

Lines 243, (Artiles, et al. [30],) 259 ( Artiles, Sanchez-Migallon Guzman, Beaufrère and Phillips [30]) and  269 (Artiles, Sanchez-Migallon Guzman, Beaufrère and Phillips [30],) - Attention to bibliography references citation

Line 307 - Is it not relevant to register the sun exposure of the accommodation or the time that the animals graze in the open air? Please relate sun exposure to vitamin D activation.

Line 331 - Animals that graze freely choose the different types of food that allow for optimal tooth wear.

Author Response

Reviewer Nº3:

We greatly appreciate your comments and suggestions, all of them were included in the new version of the manuscript.

Dear authors,

The study, as mentioned in the text, is related to a certain region of Chile, which in my opinion should be referred to in the title and more explicitly in the text, given that the dissemination of the article after publication will be worldwide and not national.

Q1.- Title: Dental disease in rabbits (Oryctolagus cuniculus)  - a private practice study in the Metropolitan Region of Chile.

R: We very much appreciate the comment. Authors agree and propose this new Title: “Dental disease in rabbits (Oryctolagus cuniculus) and its risk factors - A private practice study in the Metropolitan Region of Chile”. Changes can be observed at lines 2 – 4.

Q2.- Lines 50 to 51 - In addition, being heterodontos, they also have molars and premolars, which make up the functional unit in chewing, called molariforms  or  cheek teeth.

R: We very much appreciate the comment. Change was incorporated at lines 58 -59.

Q3.- Line 128 - 2.4 Data Collection

R: We very much appreciate the comment. Author changed the name of the subsection to “2.4 Data collection” at line 152.

Q4.-Line 132 -  age1 (measured_

R: We very much appreciate the comment. Change was incorporated at line 154.

Q5.- Do you have the bibliographic reference of the body condition scale?

R: We very much appreciate the comment. Reference was added at line 157 (https://doi.org/10.1111/jsap.12301)

Q6.- statistical p in italics

R: We very much appreciate the comment. Changes were incorporated at lines 205, 211, and 269 – 296.

Q7.- Line 185 - A mean age of 31 months was obtained, in which 30% were between neonates and juveniles, 64% were adults, and 6% corresponded to the elderly group.

Authors must previously, in the material and methods section, refer to the considered age groups.

R: We very much appreciate the comment. Authors agree with the comment and changes were incorporated at lines 154 – 157.

“age was later categorized in newborn (up to 3 months), juveniles (from 3 to 12 months), adults (from 12 months to 7 years) and senior (more than 7 years) considering prior information (https://doi.org/10.1515/jbcpp-2018-0002)”

Q8.- Line 233 and 301 - prevalence and incidence are not synonymous. In this study, prevalence was studied, not incidence.

R: We very much appreciate the comment. Change was incorporated at line 396.

Q9.- Lines 243, (Artiles, et al. [30],) 259 ( Artiles, Sanchez-Migallon Guzman, Beaufrère and Phillips [30]) and  269 (Artiles, Sanchez-Migallon Guzman, Beaufrère and Phillips [30],) - Attention to bibliography references citation

R: We very much appreciate the comment. Change was incorporated at lines 316, 340 and 349. It is important to mention that the MDPI citation style indicates that when there are more than three authors, the first appearance in the document is written with et al., while the successive ones display the complete list of authors.

Q10.- Line 307 - Is it not relevant to register the sun exposure of the accommodation or the time that the animals graze in the open air? Please relate sun exposure to vitamin D activation.

R: We very much appreciate the comment. Authors agree, it is very important to record this information. However, this data was not recorded in the patients' clinical records and the tutors provide unreliable information on these aspects, increasing the probability of information bias. An additional sentence was added to the manuscript at lines 421 - 425

“In turn of, it’s important to highlight that sun exposure, ultraviolet B (UVB) radiation, plays an important role in the synthesis and activation of vitamin D, when animals do not acquire it through their diet, potentially leading to dental alteration in animals with indoor lifestyle without sunlight or UVB exposure  (https://doi.org/10.1136/vr.145.16.452; https://doi.org/10.1053/j.jepm.2018.04.016; http://urn.fi/URN:ISBN:978-951-51-6559-6)”

Q11.- Line 331 - Animals that graze freely choose the different types of food that allow for optimal tooth wear. OK

R: We very much appreciate the comment. Authors agree with this information, changes were added at lines 420 – 421

“(…) , also animals  that graze freely, choose different types of food that will allow for optimal tooth wear (…)”

Round 2

Reviewer 1 Report

The revision of text and table are fine.

The response to reviewers is satisfy.

Reviewer 3 Report

Dear authors,

The work in its current version is more robust, which will prove to be an added value for the team.